# Associated Effects of Cadmium and Copper Alter the Heavy Metals Uptake by *Melissa Officinalis*

**DOI:** 10.3390/molecules24132458

**Published:** 2019-07-04

**Authors:** Dorota Adamczyk-Szabela, Katarzyna Lisowska, Zdzisława Romanowska-Duda, Wojciech M. Wolf

**Affiliations:** 1Institute of General and Ecological Chemistry, Lodz University of Technology, 90-924 Lodz, Zeromskiego 116, Poland; 2Department of Plant Ecophysiology, Faculty of Biology and Environmental Protection, University of Lodz, 90-237 Lodz, Banacha 12/16, Poland

**Keywords:** *Melissa officinalis*, herbs, heavy metals interactions, photosynthesis indicators, HR-CS FAAS

## Abstract

Lemon balm (*Melissa officinalis*) is a popular herb widely used in medicine. It is often cultivated in soils with substantial heavy metal content. Here we investigate the associated effects of cadmium and copper on the plant growth parameters augmented by the manganese, zinc, and lead uptake indicators. The concentration of all elements in soil and plants was determined by the HR-CS FAAS with the ContrAA 300 Analytik Jena spectrometer. Bioavailable and total forms calculated for all examined metals were augmented by the soil analyses. The index of chlorophyll content in leaves, the activity of net photosynthesis, stomatal conductance, transpiration rate, and intercellular concentration of CO_2_ were also investigated. Either Cd or Cu acting alone at high concentrations in soil are toxic to plants as indicated by chlorophyll indices and gas exchange parameters. Surprisingly, this effect was not observed when both metals were administered together. The sole cadmium or copper supplementations hampered the plant’s growth, lowered the leaf area, and altered the plant’s stem elongation. Analysis of variance showed that cadmium and copper treatments of lemon balm significantly influenced manganese, lead, and zinc concentration in roots and above ground parts.

## 1. Introduction

Plants utilize diverse strategies for the uptake of heavy metals from soil. This issue has been thoroughly investigated over years yielding a continuously growing number of publications [1,2]. One of the most important questions is related to the way in which particular metals enter and further migrate in the plant body. Several mechanisms and strategies have been reported [3]. However, we should bear in mind that real soil is a complex matrix in which components promptly interact with one another [4]. This issue is of particular relevance when heavy metals uptake and further migration are concerned. In this study we describe the combined impact of copper and cadmium on manganese, lead, and zinc uptake by the lemon balm (*Melissa officinalis L.*). 

Despite remarkable progress in modern medicinal chemistry herbal therapies, almost 80% of the world’s population relies on herbal medicine [5]. Therefore, herbs are important commodities on the global market. In particular, the production of dry herbal raw material in Poland approaches 20,000 tons annually, giving this country a leading position in Europe [6]. *Melissa officinalis*, also known as lemon balm, honey balm, common balm or balm mint, is a perennial herbaceous plant in the mint family *Lamiaceae*. This plant is native to south-central Europe, the Mediterranean Basin, Iran, and Central Asia, but now has been naturalized in both Americas and elsewhere [7]. It is widely used in medicine all over the world. Its leaves contain flavonoids, beneficial volatile compounds, triterpenes, and polyphenols. *M. officinalis* possesses sedative, antibacterial, antiviral, and antifungal activities [8]. It is not demanding and is an easy-to-grow herb, which can be cultivated in diverse soils and climatic conditions. The perspective applications of either extracts or essential oils may be related to the antioxidant activities as confirmed by several recent publications [9]. In view of those reports, lemon balm may be a useful source of rosmarinic acid and other phenolic compounds. According to Dastmalchi et al. [10], the former is responsible for the anti- acetylcholinesterase activity of *M. officinalis* extracts and may be applied in controlling Alzheimer’s disease [11]. Those extracts were also administered in multidimensional cancer therapies [12].

The diverse conditions of lemon balm cultivation inflict changes in the constitution of this particular plant and affect its medical value [13]. Furthermore, the latter may be also affected by the processing method and storage conditions of the harvested herbs [14]. Especially, heavy metals in soil and their either uptake or accumulation by plants should be carefully controlled. The EU regulations as applied to herbs are restricted to toxic elements, i.e., cadmium, lead, and chromium only [15,16]. Unfortunately, metals widely regarded as beneficial may also be stressful to plants at ambient concentrations.

Cadmium and copper are classified as priority pollutants. Copper is essential for the plant’s growth. However, its elevated concentrations in soil may lead to toxicity symptoms and hamper proper plant development [17]. Cadmium is a toxic element. Indeed, our recent study showed that its supplementation to soil reduced the growth of the lemon balm (*Melissa officinalis L.*) plant and decreased all relevant photosynthesis indicators [18]. Copper and cadmium associated interactions have been scarcely studied so far. To the best of our knowledge, their impact on the uptake of heavy metals widely distributed in the soil environment has not been investigated yet. Copper and cadmium are reported to enter the plant cell through either a competing or synergistic way, respectively [19]. Verification of this hypothesis for plants cultivated in real soil environments is of practical relevance when migrations of metals originating from wastes, sewage, or fertilizers are concerned. 

In this publication we show the influence of copper and cadmium on manganese, lead, and zinc uptake by lemon balm (*Melissa officinalis L.*). Both former metals are prone to associated interactions. Nevertheless, their combined impact on heavy metals uptake by plants has not been investigated so far. 

## 2. Results and Discussion

The investigated soil was slightly acidic (pH = 6.1) with the organic matter content at the 26% level. This is typical for the organic arable lands widely encountered in central Poland. The Mn, Cu, Zn, Cd, Pb bioavailable (89.2 ± 4; 2.78 ± 0.04; 19.6 ± 0.8; 0.14 ± 0.01; 8.70 ± 0.30 µg g^−1^, respectively) and total (143 ± 3; 5.54 ± 0.4; 46.4 ± 1.9; 0.21 ± 0.02; 12.7 ± 0.3 µg g^−1^, respectively) forms were below limits as specified in international regulations [20,21]. Metals content in the lemon balm cultivated in soil treated with Cd, Cu, or mixtures of both metals are presented in Figure 1. 

*Melissa officinalis* cultivated in the untreated reference soil accumulated all investigated metals mostly in roots. The important exception was manganese, which to a large extent migrated to the above ground parts of the plant. The copper treatment at either 20 µg g^−1^ or 100 µg g^−1^ had a significant impact on the Mn, Pb, Cd, and Zn content in roots only. The cadmium supplementation (2 and 8 µg g^−1^) reduced Mn, Pb, Cu, and Zn uptake by the above ground part of the *Melissa officinalis* plant as compared to that in the control sample. Surprisingly, Cd uptake by either roots or above ground parts of lemon balm was inversely proportional to the Cu concentration in soil. However, the Cu concentration in plant tissues was not directly related to the Cd content in soil.

The chlorophyll content in leaves, the activity of net photosynthesis, stomatal conductance transpiration rate, and intercellular CO_2_ concentration (Figure 2) indicate that lemon balm exhibited diverse photosynthesis activities. Either Cd or Cu acting alone at high concentrations in soil (8 and 20 µg g^−1^, respectively) are toxic to plants as indicated by chlorophyll indices and gas exchange parameters. Surprisingly, this effect was not observed when both metals were administered together. The sole cadmium or copper supplementations hampered the plant’s growth, lowered the leaf area, and altered the plant’s stem elongation. The important exception was the lowest 2 µg g^−1^ Cd dose, which increased the leaf area and the stem length (Figure 3) while at the same time prompted the leaves’ green colour to fade. The inhibition of chlorophyll synthesis upon the Cd administration in soil has been recognized and it is sometimes related to hampering the Mg uptake by particular plants [22]. Remarkably, the higher Cd doses did not affect either the plant’s growth or the leaf area as compared to the control species. It is generally accepted that Cd alters physiological processes in leaves [23]. The majority of reports emphasize the negative effects leading to the biomass decline. However, contradictory statements do also exist. In particular, Piršelová et al. [24] showed that in the faba bean plant (*Vicia faba* cv. Aštar) Cd administration to soil resulted in the decrease of either fresh or dry root weight, but at the same time it prompted the increase of shoots fresh biomass. A similar rise of fresh shoots weight was also observed by Shah et al. [25] in shisham (*Dalbergia sissoo Roxb*.). The Cd toxicity in plants has been a subject of numerous investigations [26,27]. Unfortunately it is species dependent and the final conclusions are often contradictory and by all means are far from clarity [28,29]. 

The two-way ANOVA (Table 1) shows that significant, combined Cd and Cu interactions are triggered by the Mn, Pb, and Zn as present in soil. Correlations between Cd and Cu in soil and plant environment are quite well recognized and documented in the scientific literature [30,31]. However, the associated interactions involving more heavy metals as present in the soil environment has not been studied as yet.

Cd and Cu may influence either metals uptake from the soil or their further migration within the plant body. The former may be analyzed by the transfer coefficient (TC) and bioaccumulation factor (BAF). They are defined as ratios of particular element concentrations in root and shoot, respectively, related to their content in the soil environment [32,33,34]. Metal distribution inside the plant body was assessed by the translocation factor (TF), which is the ratio of element concentration in the above ground part of the plant to that in roots [35,36,37].

The TC calculated for lemon balm plants cultivated in the reference, untreated soil was in the series Cd > Zn > Cu > Pb > Mn. The TF for the untreated soil followed the order Mn > Cu > Pb > Zn > Cd. Increase of Cd content in soil from 2 to 8 µg g^−1^ changed the position of Cu, Zn, and Pb in both series, giving the orders Cd > Cu > Zn > Mn > Pb and Mn > Zn > Cu > Pb > Cd, respectively. The BAF calculated for plants cultivated in the untreated soil was in the order Cd > Cu > Zn > Mn > Pb, which was significantly altered upon the Cd and Cu administration.

The uptake and transport of ballast metals Cd and Pb take place on a competitive basis with micro- and macroelements for trans-membrane carriers characterized by a broad specificity. Upon ion deficit in the cell, those transporters are synthesized and further activated in biological membranes. As non-specific carriers, they also transport ballast elements in excess. Cadmium entering root cells probably uses broad-spectrum transporters for copper and zinc. Therefore, the uptake of this metal in the presence of increased doses of cadmium decreases.

## 3. Materials and Methods 

### 3.1. Soil Analysis

Soil samples were collected in May 2017 from the topsoil according to the standard procedure [38] on agricultural land located at 51°22′ N, 19°49′ E (Włodzimierzów village, Piotrków Trybunalski district, Poland). All samples were dried and sifted (<2 mm). Soil pH was measured in 1 mol L^−1^ KCl solution by the potentiometric method [39]. The gravimetric method for the determination of soil organic matter by the mass loss at 550 °C was applied [40,41]. The bioavailable forms of metals were determined in 0.5 mol∙L^−1^ HCl solutions [42]. The total metal contents were measured in samples mineralized with the Multiwave 3000 instrument (Anton Paar GmbH, Graz, Austria). A mixture of concentrated HNO_3_ (6 mL) and HCl (2 mL) was applied. Metal concentrations were determined by the HR-CS FAAS with the ContrAA 300 (Analytik Jena spectrometer, Jena, Germany).

### 3.2. Preparation of Plant Material

Lemon balm was cultivated under laboratory conditions by the pot method [43]. A single pot contained 200 g of soil. The first series consisted of five pots and was cultivated as a reference without metals addition. The subsequent eight series (five samples each) were augmented with Cd(NO_3_)_2_ or Cu(NO_3_)_2_ solutions to the final metal concentrations in soil: (I) 2 µg g^−1^ Cd; (II) 8 µg g^−1^ Cd; (III) 20 µg g^−1^ Cu; (IV) 100 µg g^−1^ Cu; (V) 2 µg g^−1^ Cd and 20 µg g^−1^ Cu; (VI) 2 µg g^−1^ Cd and 100 µg g^−1^ Cu; (VII) 8 µg g^−1^ Cd and 20 µg g^−1^ Cu; (VIII) 8 µg g^−1^ Cd and 100 µg g^−1^ Cu. Seeds of *Melissa officinalis* (P.H. Legutko company, Jutrosin, Poland) were sown in an amount of 0.1 g (approximately 80 seeds) per pot. All pots were kept in a growth chamber at controlled temperatures 23 °C ± 2 °C (day) and 16 °C ± 2 °C (night). The relative humidity was limited to 70–75% while the photosynthetic active radiation (PAR) during the 16 h photoperiod was restricted to 400 µmol m^−2^ s^−1^. All plants were regularly watered by demineralized water. After three months, the above ground parts of the plants were cut, and the roots were separated from the soil and washed with demineralized water. The entire harvest was dried at 45 °C, homogenized, and grounded.

### 3.3. Determination of Metals in Basil

The dried lemon balm samples (0.5 g) were mineralized in concentrated HNO_3_ (6 mL) and HCl (1 mL) acid solutions with a microwave (Anton Paar Multiwave 3000). Metal contents were determined by the HR-CS FAAS with the ContrAA 300 Analytik Jena spectrometer. The certified reference material INCT-MPH-2 (a mixture of selected Polish herbs) was used for the validation of the analytical methodology (Table 2) [44].

### 3.4. Melissa Plant Growth and Its Physiological Activity

Plant height was measured from the soil surface up to the highest part of the leaf. Index of chlorophyll content was evaluated using the Konica Minolta SPAD-502, Tokyo, Japan, a methodology in which the chlorophyll concentration is determined by measuring the leaf absorbance in the red and near-infrared regions. Gas exchange (activity of net photosynthesis, stomatal conductance, intercellular concentration of carbon dioxide, and transpiration) was determined with the gas analyzer apparatus TPS-2 (Portable Photosynthesis System, Amesbury, MA, USA) [45,46,47,48]. All measurements were made in triplicate on separate plants.

### 3.5. Data Analysis

All analyses were repeated five times. Bartlett’s and Hartley’s tests were used to confirm the equality of investigated population variance (STATISTICA, version 10 PL, package). The normality of the data was tested using the Shapiro-Wilk test [49,50]. Two-way ANOVA was used to evaluate the combined effect of the cadmium or copper supplementation in soil on the accumulation of manganese, lead, and zinc by the lemon balm plant. All calculations were performed at the 0.95 probability level.

## 4. Conclusions

Our results showed that additive cadmium–copper interactions modified manganese, lead, and zinc uptake by *Melissa officinalis.* This issue is of particular importance when herbs are grown in soils with diverse and not fully controlled heavy metals concentrations. Either Cd or Cu acting alone at high concentrations in soil (8 and 20 µg g^−1^, respectively) is toxic to plants as indicated by chlorophyll indices and gas exchange parameters. Surprisingly, this effect was not observed when both metals were administered together. The maximum permissible concentrations (MPC) commonly used in agriculture and environmental protection assessment strategies usually accounts for single-metal toxicities and do not acknowledge additive effects. This issue has not been fully recognized by the environmental protection bodies at either national or European levels. 

## Figures and Tables

**Figure 1 molecules-24-02458-f001:**
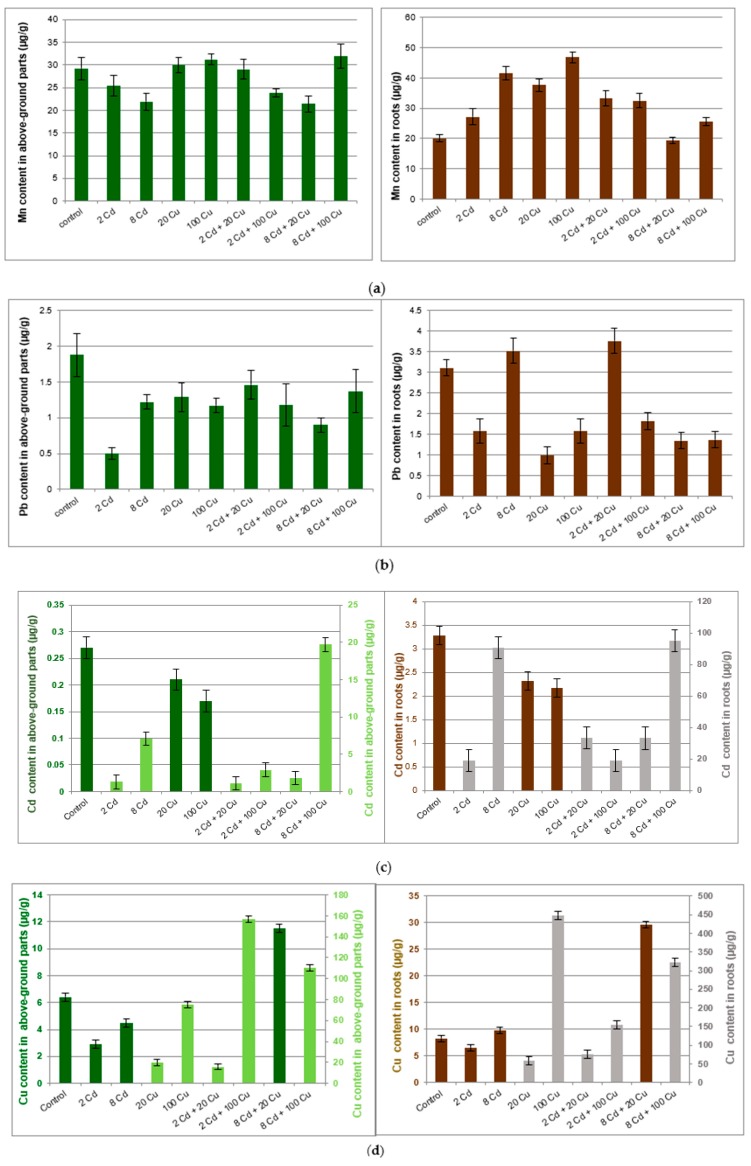
Manganese (**a**); lead (**b**); copper (**c**); cadmium (**d**); zinc (**e**) content (µg/g) in above ground parts and roots of the *Melissa officinalis* plant displayed against the Cd and Cu doses as used for the soil supplementation. 2 Cd = 2 µg/g cadmium; 8 Cd = 8 µg/g cadmium; 20 Cu = 20 µg/g copper; 100 Cu = 100 µg/g copper. In (**c**) and (**d**) dark bars show lower metal concentrations and are related to the left scale axis while pale bars (higher concentrations) are represented by the right axis.

**Figure 2 molecules-24-02458-f002:**
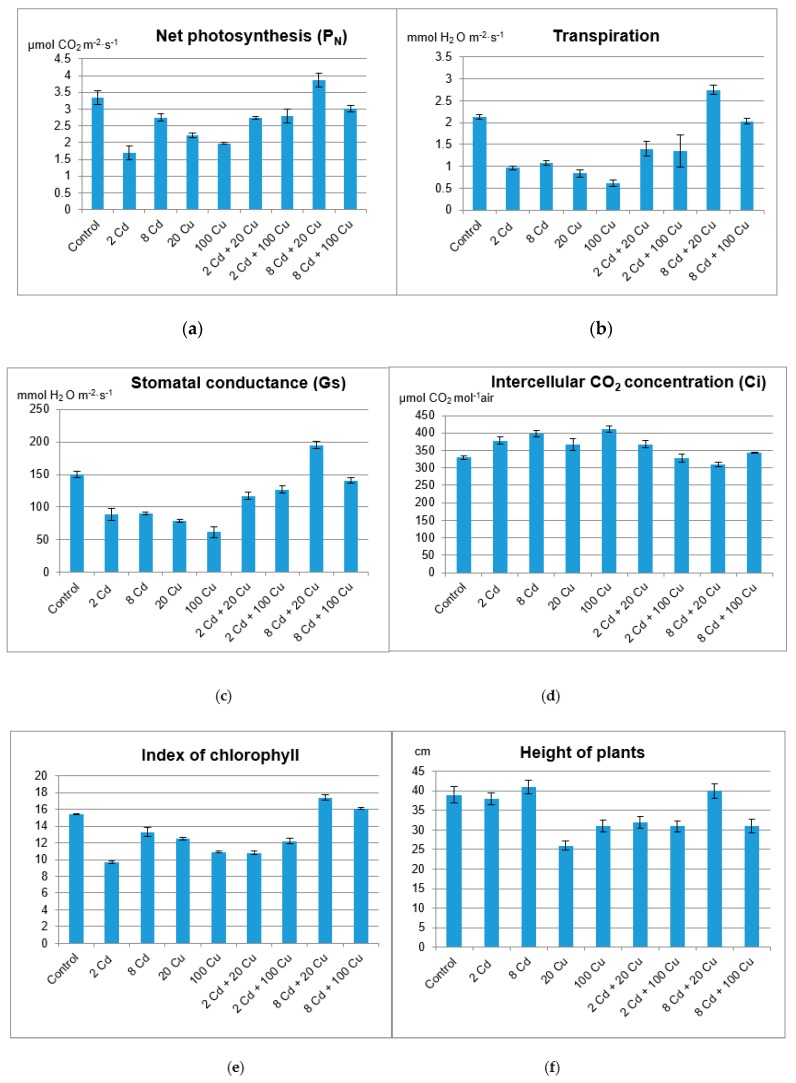
Net photosynthesis (**a**); transpiration (**b**); stomatal conductance (**c**); intercellular CO_2_ concentration (**d**); index of chlorophyll content (**e**); height of the plant (**f**) for lemon balm grown on soil either with or without Cd and Cu treatment. 2 Cd = 2 µg/g cadmium; 8 Cd = 8 µg/g cadmium; 20 Cu = 20 µg/g copper; 100 Cu = 100 µg/g copper.

**Figure 3 molecules-24-02458-f003:**
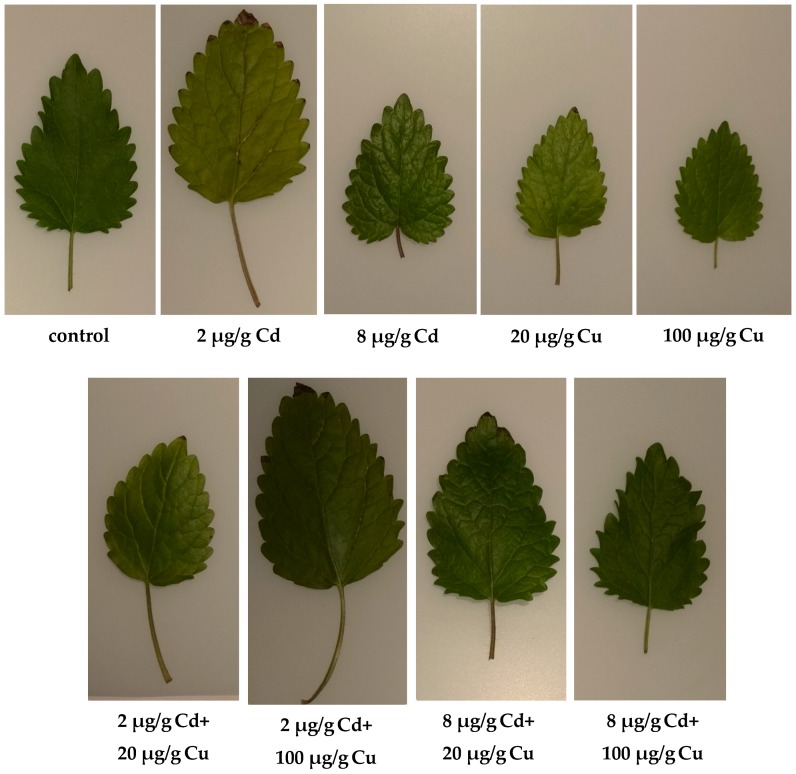
Representative leaves of *Melissa officinalis* cultivated in soils either with or without Cd and Cu treatment.

**Table 1 molecules-24-02458-t001:** Two-way ANOVA parameters* for Mn, Pb, and Zn contents in above ground parts and in roots of *Melissa officinalis* across eight soil supplementations as described in the Materials and Methods section.

**Roots**						
**Source of variation**	**SS**	**df**	**MS**	**F**	**p**	**Test F**
Samples	7961.42	8	995.18	207.72	6.35 × 10^−62^	2.0252
Metals	96,997.42	2	48,498.71	10,123.28	1.40 × 10^−123^	3.0803
Interactions	9687.58	16	605.47	126.38	3.59 × 10^−62^	1.7380
**Above ground parts**						
**Source of variation**	**SS**	**df**	**MS**	**F**	**p**	**Test F**
Samples	2514.20	8	314.28	106.00	1.49 × 10^−47^	2.0252
Metals	36,578.72	2	18,289.36	6168.95	4.70 × 10^−112^	3.0803
Interactions	2080.99	16	130.06	43.87	9.17 × 10^−40^	1.7380

*SS—sum of squares; df—degrees of freedom; MS—mean square; p—probability value; F—calculated Snedecor’s F parameter; Test F—Snedecor’s F critical value.

**Table 2 molecules-24-02458-t002:** Metals concentration in the certified reference material (*p* = 0.95; *n* = 6).

Metal	Certified Value µg g^−1^	Found µg g^−1^	Recovery %
**Manganese**	191 ± 12	188 ± 8	98
**Lead**	2.16 ± 0.23	2.13 ± 0.13	98
**Copper**	7.77 ± 0.53	7.50 ± 0.38	96
**Cadmium**	0.199 ± 0.015	0.206 ± 0.007	103
**Zinc**	33.5 ± 2.1	34.2 ± 0.7	102

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
