# Peer review of "Associated Effects of Cadmium and Copper Alter the Heavy Metals Uptake by Melissa Officinalis"

_molecules, 2019, doi:10.3390/molecules24132458_

Round 1

Reviewer 1 Report

The work under review covers large information on the relation of the influence of copper and cadmium on manganese, lead and zinc uptake by the lemon balm based on the results of the trace elements determination using modern equipment. The measurements and data interpretation and application seem to have been done well, also the trace elements contents in above-ground parts and in roots of Melissa officinalis across eight soils supplementations. So, the paper adds valuable new information concerning our knowledge on the biochemistry of lemon balm, which is affected by the content of cadmium and copper in the roots and above-ground components. The description is complete and well presented. The paper can be recommended for publication in Molecules with some minor revisions.

 Comments (the number is the line code in the pdf file): they are intended to improve a future paper, which must be focused on the statistical data analysis.

(1) Sections of the article should be swapped: first 2. Materials and Methods; and then 

3. Results and Discussion.

(2) It is more clear to write: values of TC instead of TCs (line 142), and  the values of BAS instead of BASs (line 145)

(3) The abbreviated names in first row of table 1 (SS, df, MS, F. p- value, Test F) should be explained in the text or in the notes to the table 1.

Author Response

Comments (in bold) to the Referee reports.  For clarity relevant fragments of the report are indicated in italic. 

Thank you very much for your helpful comments. All suggested improvements were incorporated into revised version of the manuscript.

(1) Sections of the article should be swapped: first 2. Materials and Methods; and then 3. Results and Discussion.

Following the Referee suggestion the order of Materials and Methods and Results and Discussion has been swapped.

(2) It is more clear to write: values of TC instead of TCs (line 142), and  the values of BAS instead of BASs (line 145)

Values of TC, BAS  has been corrected following the Referee suggestions

(3) The abbreviated names in first row of table 1 (SS, df, MS, F. p- value, Test F) should be explained in the text or in the notes to the table 1.

We followed the usual practices of ANOVA results presentations. ANOVA parameters are defined in numerous books and publications with the generally accepted symbols. Following the Referee suggestion a short description of parameters listed in the Table 2 has been added at the bottom of the table.

Reviewer 2 Report

The manuscript MOLECULES536731 deals with the associated effect of Cd and Cu on the metal uptake (Mn, Zn and Pb) by the lemon balm (Melissa officinalis L.).

The paper is well written and data are well presented. The results are very interesting and I suggest the paper for publication, after very few minor revisions. My comments are as follows:

I suggest to move the methodology section before the Results and discussion session, since to avoid to go up and down during the reading.

Graphs are very poor. Many of them are showed without X and Y-axis title. I suggest to improve the several graphs format.

Did the authors computed the LOD for the metals measured by FAAS. I suggest to compute the LOD according to Harmonized Tripartite Guideline, Validation of Analytical Procedure (see Truzzi et al., Analytical Letters, 47 (2014) 1118-1133).

Author Response

Comments (in bold) to the Referee report.  For clarity relevant fragments of the report are indicated in italic. 

Thank you very much for your helpful comments. All suggested improvements were incorporated into revised version of the manuscript.

 I suggest to move the methodology section before the Results and discussion session, since to avoid to go up and down during the reading.

Following the Referee suggestion the order of Materials and Methods and Results and Discussion has been swapped.

Graphs are very poor. Many of them are showed without X and Y-axis title. I suggest to improve the several graphs format.

We particularly grateful for this comment. Indeed, graphs as presented in the original manuscripts were not clear. Improved figures with specially adjusted scales were included in the revised manuscript.

 Did the authors computed the LOD for the metals measured by FAAS. I suggest to compute the LOD according to Harmonized Tripartite Guideline, Validation of Analytical Procedure (see Truzzi et al., Analytical Letters, 47 (2014) 1118-1133).

We determined heavy metals in plants by high resolution, continuous source  FAAS instrument. We use this spectrometer since 2017 and initially determined LOD’s for metals we use to investigate.  Limits of detection for manganese, zinc, lead, cadmium and copper were 1.1; 0.9; 10.0; 0.5; 0.6 µg/L respectively. These values are quite comparable to data reported for various FAAS spectrometers. Metal concentrations as determined by us in Lemon balm are a few orders of magnitude higher.

Reviewer 3 Report

Dear authors

please find in attached file my remarks. For me, this study is very interesting, well described and the results are serious and consistent.That's why I suggest publication of the current study in the Molécules Journal.

Author Response

Thank you very much for your helpful comments.

In line 168 is:

The subsequent eight series (five samples each) were augmented with Cd(NO3)2 and Cu(NO3)2 solutions to the final metal concentrations in soil (I) 2 μg g-1 Cd; (II) 8 μg g-1 Cd; (III) 20 μg g-1 Cu; (IV) 100 μg g-1 Cu; (V) 2 μg g-1 Cd and 20 μg g-1 Cu; (VI) 2 μg g-1 Cd and 100 μg g-1 Cu; (VII) 8 μg g-1  Cd and 20 μg g-1 Cu; (VIII) 8 μg g-1 Cd and 100 μg g-1 Cu.

We understand that the Referee pointed out the ambiguity in the sentence describing usage of cadmium and copper nitrates for the plant treatments. To make the respective sentence more clear we replaced “and” with “or” because both nitrates were used either solely or in the combined forms.